# Variations in Levels and Sources of Atmospheric VOCs during the Continuous Haze and Non-Haze Episodes in the Urban Area of Beijing: A Case Study in Spring of 2019

**Lihui Zhang** [1], **Xuezhong Wang** [1], **Hong Li** [1], **Nianliang Cheng** [1,2], **Yujie Zhang** [1,*], **Kai Zhang** [1] **and Lei Li** [3]

[1] State Key Laboratory of Environmental Criteria and Risk Assessment, Chinese Research Academy of Environmental Sciences, Beijing 100012, China; zlhlihui@126.com (L.Z.); wangxz@craes.org.cn (X.W.); lihong@craes.org.cn (H.L.); cnl88@163.com (N.C.); zhangkai@craes.org.cn (K.Z.)
[2] Beijing Municipal Environmental Monitoring Center, Beijing 100048, China
[3] Academy of Environmental Planning & Design, Co., Ltd., Nanjing University, Nanjing 210093, China; lilei@njuae.cn
[*] Correspondence: zhangyj@craes.org.cn; Tel.: +86-10-8493-1717

**Abstract:** To better evaluate the variations in concentration characteristics and source contributions of atmospheric volatile organic compounds (VOCs) during continuous haze days and non-haze days, hourly observations of atmospheric VOCs were conducted using a continuous on-line GC-FID (Airmo VOC GC-866) monitoring system during 1–15 March 2019, in urban areas of Beijing, China. The results showed that the total VOC concentrations during haze days and non-haze days were $59.13 \pm 31.08$ μg/m$^3$ and $16.91 \pm 7.19$ μg/m$^3$, respectively. However, the average O$_3$ concentrations during the two haze days were lower than those of non-haze days due to the extremely low concentrations at night instead of the reported lower photochemical reaction in daytime. The ratio of OH radical concentration during haze and non-haze days indicating that the rate of photochemical reaction during haze days was higher than those of non-haze days from 13:00–19:00. The stable air conditions and the local diesel emission at night were the main reasons for the decreased O$_3$ concentrations during haze days. Six major sources were identified by positive matrix factorization (PMF), namely, diesel exhaust, combustion, gasoline evaporation, solvent usage, gasoline exhaust, and the petrochemical industry, contributing 9.93%, 25.29%, 3.90%, 16.88%, 35.59% and 8.41%, respectively, during the whole observation period. The contributions of diesel exhaust and the petrochemical industry emissions decreased from 26.14% and 6.43% during non-haze days to 13.70% and 2.57%, respectively, during haze days. These reductions were mainly ascribed to the emergency measures that the government implemented during haze days. In contrast, the contributions of gasoline exhaust increased from 34.92% during non-haze days to 48.77% during haze days. The ratio of specific VOC species and PMF both showed that the contributions of gasoline exhaust emission increased during haze days. The backward trajectories, potential source contribution function (PSCF) and concentration weighted trajectory (CWT) showed that the air mass of VOCs during haze days was mainly affected by the short-distance transportation from the southwestern of Hebei province. However, the air mass of VOCs during non-haze days was mainly affected by the long-distance transportation from the northwest.

**Keywords:** atmospheric VOCs; haze and non-haze; source apportionment; potential source contribution function; concentration weighted trajectory

## 1. Introduction

Haze pollution, which is characterized by a high concentration of PM$_{2.5}$ (fine particulate matter with an aerodynamic diameter equal to or less than 2.5 μm), significantly affects atmospheric visibility [1,2], air quality, and global climate change [3], and is associated with increased respiratory symptoms and deaths [4,5]. The rapid development

of the economy, involving industrialization and urbanization, has triggered numerous environmental pollution problems in China, and, in particular, haze pollution. In 2016, among 338 prefecture-level and higher cities, 254 cities (75.1%) did not meet the China Air Quality Standard [6]. The average annual concentration of $PM_{2.5}$ of Beijing reached 73 $\mu g/m^3$ in 2016. This value is 1.09 times above the corresponding secondary standard limits specified in the Ambient Air Quality Standard [7]. The average annual concentration of Beijing decreased to 51 $\mu g/m^3$ in 2018 [8] and the number of haze days noticeably declined due to the implementation of the "2 + 26" urban joint prevention and the various emission control measures, such as upgrading low-efficiency coal-fired industrial boilers and furnaces, tightening vehicle emission standards, controlling VOCs emission in the spraying industry and coal-fired power plants, and carrying out oil and gas recovery in gas stations. However, more effort must be made to meet the national annual standard of 35 $\mu g/m^3$.

Although a significant amount of research has been conducted on haze, most studies have focused on measurements of particle matter to reveal the deterioration of air quality during haze days [3,4]. However, the formation mechanism of haze not only involves its physical evolution, but also chemical reactions [9–11]. Secondary organic aerosol (SOA) is an important component of $PM_{2.5}$ [11–13]. According to recent studies, serious haze events are driven to a significant degree by intensive formation of secondary aerosols, and aggravated by unfavorable meteorological conditions (wind, precipitation, the planetary boundary layer (PBL), etc.) [14–19]. Huang estimated a fraction of 55–77% SOA to $PM_{2.5}$ during serious haze pollution [12]. Sun found that when visibility decreased from 50 to 1 km, SOA increased from 2.1 to 13.2 $\mu g/m^3$ [17]. Volatile organic compounds (VOCs) are important precursors of SOA. VOCs chemically react with oxidants ($O_3$, $HO_2$ radicals, and OH radicals) in the atmosphere to produce SOA, which then undergoes a series of photochemical reactions or physical evolutions to form haze pollution. Thus, it is important to conduct research on VOCs during haze days. However, studies undertaken to date of VOCs during haze days are relatively few. Sun researched VOC characteristics during a typical hazy episode in Beijing in January 2013 and found that aromatics were the dominant contributors to SOA formation, accounting for 56.3% during non-haze days and up to 85.7% during haze days [17]. Wei conducted a comparison study of VOCs between haze and non-haze days in July and December of 2015, and found a significant VOC chemical degradation during summer haze days, but the degradation in winter was not explicit [18].

Therefore, this study conducted a comparison study of VOCs levels and sources between haze days and non-haze days during continuous haze episodes in Beijing. The hourly concentrations of 59 VOCs were measured using an Airmo VOC online analyzer during 1–15 March 2019. In addition, regional transport pathways of VOCs were identified by the potential source contribution function (PSCF) model and concentration-weighted trajectory (CWT) model.

## 2. Materials and Methods

### 2.1. Observation Site and Period

The observation site is located on the second floor of the Atmospheric Photochemical Smog Simulation Laboratory in the Chinese Research Academy of Environmental Sciences (CRAES) in the Chaoyang District of Beijing (40.04° N, 116.42° E). The sampling port is 8 m above ground. The site is 2 km from the North Fifth Ring Road and is about 3.6 km from the Beijing Olympic park. The north-to-south Beijing Subway Line 5 and Beiyuan Road lie about 200 m to the west; the east-to-west Chunhua Road lies about 100 m to the south; and the Beijing Subway Line 13 lies about 700 m to the north-east. Detailed information about the observation site was provided in our previous studies [20–22].

During the whole observation period, 8 and 7 days were classified as haze days (AQI > 100) and non-haze days (AQI ≤ 100), respectively. The $PM_{2.5}$ average concentrations corresponding to AQI > 100 and AQI ≤ 100 were 114.37 $\mu g/m^3$ and 11.42 $\mu g/m^3$,

respectively. Among these observing days, haze days were from 1 March 2019 to 5 March 2019 and from 8 March 2019 to 10 March 2019, and the remainder were non-haze days.

### 2.2. Observing Instruments

VOCs were continuously observed and analyzed in ambient air using an Airmo VOC online analyzer (GC-866, Chromatotec Group, France). Detailed information about the observation instrument was provided in our previous studies [20–22]. External calibration correlations ($R^2$) were more than 0.9. The instrument was internally calibrated once per 24 h, using n-butane, n-hexane, and benzene, such that the deviation should not be more than 10%.

In this study, concentrations of $PM_{2.5}$, CO, $O_3$, and NO*x* were all monitored by instruments from Thermo Environmental Instruments (Thermo Scientific, America). Detailed information about the observation instruments was provided in our previous studies [20].

### 2.3. Methods

#### 2.3.1. Positive Matrix Factorization

Positive matrix factorization (PMF) is one of the most commonly used receptor models. It is a mathematical approach for analyzing and quantifying the contribution of sources to samples based on the chemical composition or the fingerprint of targeted sources [9,23–27]. Two input files are required in the PMF model: the concentration file and the uncertainty file. Before implementing PMF, the data files were processed separately.

For the concentration file:

The concentration is the measured value when it is greater than the minimum detection limit (MDL). When the concentration is less than the MDL, the concentration is half of the MDL. The concentration was determined using Equation (1):

$$Con = \begin{cases} \text{the measured value, } (Con \geq \text{MDL}) \\ \frac{1}{2} \times \text{MDL, } (Con < \text{MDL}) \end{cases} \tag{1}$$

where *Con* is concentration, MDL is the minimum detection limit.

For the uncertainty file:

The uncertainty was determined using Equation (2):

$$Un = \begin{cases} \sqrt{\text{EF} \times \text{Con}^2 + (0.5 \times \text{MDL})^2}, (Con \geq \text{MDL}) \\ 5/6 \times \text{MDL}, (Con < \text{MDL}) \end{cases} \tag{2}$$

where *Un* is uncertainty, EF is error fraction.

The range of error fraction (EF) is defined as follows: when the amount of observed data are relatively abundant and the value of concentration is higher than the MDL, EF is 0.05; when the observed results are composed by a relatively large proportion of zeros, EF could be set to any value in the range of 0.2 to 0.3 [23]. EF in this study was 0.2.

The PMF 5.0 model was applied in this study to identify the sources during the whole observation period and different pollution events. Thirty-five species were selected from 59 VOC species and input into the model for calculation (https://www.epa.gov/air-research/positive-matrix-factorization-model-environmental-data-analyses).

#### 2.3.2. OH Radical Concentration

In the urban areas, ethylene (ET) and acetylene (A) are mainly derived from incomplete combustion processes of fossil fuels (such as vehicle exhaust and coal combustion) [28], and their reactivity differs by nearly an order of magnitude. The change in the ET/A ratio mainly reflects the difference in photochemical reaction intensity. Based on the first order kinetic reaction of NMHCs with OH radicals, the relationship between ET/A and OH radical concentration can be deduced.

ET/A was determined using Equation (3):

$$ET/A = ERe^{-(k_{ET}-k_A)[OH]\Delta t}$$ (3)

where ET and A represent ethylene and acetylene concentration, respectively; ER is the emission ratio of ethylene to acetylene and its value is 1.80 [29]; $k_{ET}$ and $k_A$ are reaction rate constants of ethylene and acetylene with OH radical, respectively; (OH) represents OH radical concentration; $\Delta t$ is reaction time, namely photochemical age.

According to Equation (3), the ratio of OH radical concentration during haze days and non-haze days was determined using Equation (4):

$$[OH]_{Haze}/[OH]_{non-Haze} = \frac{\ln(ER) - \ln[(ET/A)]_{Haze}}{\ln(ER) - \ln[(ET/A)]_{non-Haze}}$$ (4)

### 2.3.3. The Backward Trajectory

The main purpose of the backward trajectory clustering is to cluster trajectories with similar geographical origins. Backward trajectory clustering based on the GIS-based software TrajStat was used in this study to understand the history of air masses [30]. The meteorological data were obtained from the Global Data Assimilation System (GDAS) dataset (http://ready.arl.noaa.gov/archives.php). This study calculated 24 h backward trajectories during haze days and non-haze days occurring at the observation site (40.04° N, 116.42° E). The arrival level was set at 500 m above ground level and the model was run every hour.

### 2.3.4. Potential Source Contribution Function (PSCF)

The PSCF is a method combining the backward trajectory and a pollution value (this study refers to VOC concentrations) for identifying potential source areas of pollution. The study field was divided into small equal grid cells (*ij*).

The PSCF value was defined using Equation (5):

$$PSCF_{ij} = \frac{m_{ij}}{n_{ij}}$$ (5)

where *i* and *j* represent latitude and longitude, $n_{ij}$ is the number of endpoints that fall in the *ij* cell, and $m_{ij}$ is the number of endpoints in the same cell that are associated with samples that exceed the threshold criterion. A higher PSCF value indicates that the region corresponding to the grid is the potential source region of high concentration pollution at the observation site, and the trajectory through this area is the transmission path with a significant influence on the observation site. Detailed information about the PSCF was provided in other relevant references [30–32].

### 2.3.5. Concentration-Weighted Trajectory (CWT)

Because PSCF only reflects the proportion of pollution trajectories in a grid, which cannot reflect the pollution levels of trajectory, CWT is used to weight trajectories with associated concentrations. The geographical domain was divided into grid cells, each covering an area of $0.5° \times 0.5°$.

The CWT was determined using Equation (6):

$$C_{ij} = \frac{\sum_{l=1}^{M} C_l \times t_{ijl}}{\sum_{l=1}^{M} t_{ijl}}$$ (6)

where $C_{ij}$ represents the average weight concentration of back trajectory *l* in the *ij* cell; $C_l$ is trajectory *l* through the *ij* cell corresponding to VOC concentration; and $t_{ijl}$ is the time that trajectory *l* stays in the *ij* cell.

To date, numerous reports have applied PSCF, CWT, and other models to identify the transport paths and potential source areas for the atmospheric particles and chemical species, by coupling observed chemical concentrations with meteorological information [30–32].

## 3. Results and Discussion

### 3.1. Time Series of VOCs and other Pollutants

Figure 1 shows the time series of observed VOCs, $PM_{2.5}$, CO, NO$x$, O$_3$, and meteorological parameters (wind speed and wind direction) from 1 March 2019 to 15 March 2019. During the whole observation period, the variation in VOCs was consistent with the variations in $PM_{2.5}$, CO, and NO$x$, which indicated that the main sources for these pollutants were similar and their change was mainly influenced by the meteorological conditions. Two haze events were observed from 1 March 2019 to 5 March 2019 and from 8 March 2019 to 10 March 2019. During the two haze events, the maximum hourly concentrations of $PM_{2.5}$ were 263.58 and 156.34 $\mu g/m^3$, respectively, and the average hourly concentrations were 142.96 and 95.33 $\mu g/m^3$, respectively, which were 7–12 times higher than the average concentrations during non-haze days (12.39 $\mu g/m^3$). The maximum hourly concentrations of VOCs were 148.53 and 103.56 $\mu g/m^3$, respectively, and the average hourly concentrations were 63.37 and 55.47 $\mu g/m^3$, respectively, which were 3–4 times higher than the average concentrations during non-haze days (16.91 $\mu g/m^3$). The average concentrations of CO during different haze events were 1.74 and 1.55 $mg/m^3$, respectively, which were evidently higher than those during non-haze days (0.82 $mg/m^3$). The average concentrations of NO$x$ were 88.71 and 108.34 $\mu g/m^3$, respectively, which were significantly higher than that of 28.01 $\mu g/m^3$ during non-haze days. The peak values of NO$x$ usually occurred at around 02:00, and the diesel vehicles are only permitted to drive on the Fifth Ring Road after 0:00 at night, and thus diesel exhaust was one of the main sources of local NO$x$. Figure 1 shows that wind speeds were lower during the two haze events, and the average wind speeds were 2.00 and 1.88 m/s, respectively, whereas the average wind speed was 3.40 m/s during non-haze days. Wind speed could influence the dilution and diffusion of pollutants. The much lower wind speeds were one of the important factors for the accumulation of pollutants during haze events.

However, the peak values of O$_3$ appearing during 12:00–15:00 in the afternoon as the product of photochemical reaction, which was different with other pollutants. Unlike the concentrations of other pollutants, which during haze days were much higher than those during non-haze days, the average O$_3$ concentrations during the two haze events declined, with the average concentrations of 64.31 and 38.30 $\mu g/m^3$ during haze days, and an average concentration of 71.51 $\mu g/m^3$ during non-haze days. However, the average maximum daily 1-h (MDA1h) O$_3$ concentrations during two haze events were both higher than those of non-haze days, which were 131.76 $\mu g/m^3$ and 102.83 $\mu g/m^3$ during two haze events and 93.61 $\mu g/m^3$ during non-haze days. The much lower average O$_3$ concentrations during the two haze events were mainly due to the extremely lower O$_3$ concentrations of the nights during haze days. It can be seen in Figure 1 that O$_3$ concentrations reached almost zero during 00:00–02:00 when the concentrations of NO$x$ were the highest during haze days, while the O$_3$ concentrations stayed high at nights of non-haze days. The night-time O$_3$ reacts with NO, NO$_2$ and alkene to form NO$_2$, NO$_3$ and OH radicals. During haze nights, the much lower PBL height and lower windspeeds result in higher NO$x$ concentrations, which reacts with O$_3$ and results in lower O$_3$ concentrations. However, during non-haze days, the NO$x$ concentrations were very low due to the much higher windspeeds, especially on 6, 7, 12, 15 March, the NO$x$ concentrations became almost zero. The much lower NO titration effect was one of the main reasons for much higher O$_3$ concentrations in nights of non-haze days. Besides, long distance transport of O$_3$ due to high windspeeds might be another important reason. Previous research suggested that the decrease in O$_3$ concentrations during haze days was due to the reduced photochemical formation caused by the scattering of high concentrations of $PM_{2.5}$ during haze events [9,10], e.g., Sheng

observed the lower photochemical reaction of VOCs during haze days than that during clear days in winter in Beijing by comparing the mixing ratios of benzene/C7-aromatics and benzene/C8-aromatics [33]. However, we did not find evident differences in the mixing ratios of benzene/C7-aromatics and benzene/C8-aromatics during the two periods. In the troposphere, VOCs are transformed by the chemical processes of photolysis (at wavelengths > 290 nm because shorter wavelengths are absorbed by $O_2$ and $O_3$ in the stratosphere), reaction with the OH radicals (typically during daylight hours), reaction with the nitrate ($NO_3$) radicals during evening and nighttime hours and reaction with $O_3$ [34]. Thus, the average diurnal variation of OH radical concentration ratios during haze days and non-haze days for the daytime were calculated using Equation (4) and illustrated in Figure 2. It can be seen that from 13:00 to 19:00, the average OH radical concentration during haze days were higher than those of non-haze days, indicating that the rate of photochemical reaction during haze days was higher than those of non-haze days. However, further investigation must be conducted as our study is only based on the results of 15 days in spring.

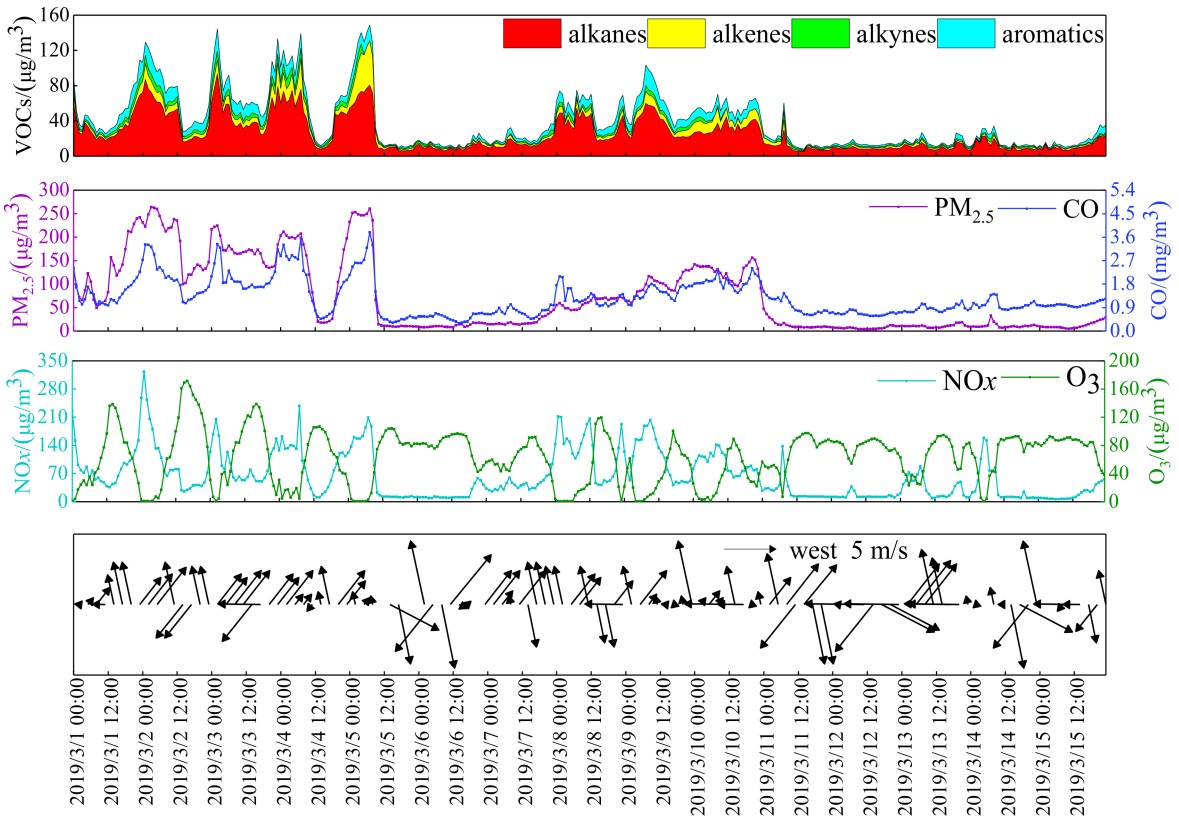

**Figure 1.** Time series of observed VOCs, $PM_{2.5}$, CO, NO*x*, $O_3$ and meteorological parameters (wind speed and wind direction).

### 3.2. Concentrations and Compositions of VOCs

The average concentrations and ranges of 59 VOCs species during different observation periods are summarized in Table 1. The concentrations of 59 VOCs varied between 7.92 and 60.30 μg/m³, with an average mass concentration of 16.91 ± 7.19 μg/m³ during non-haze days. However, this rose to 59.13 ± 31.08 μg/m³ during the haze events, and the average mass concentration was 3.5 times larger than that during non-haze days. Alkanes were the most abundant components, accounting for 61.54% (36.39 ± 18.64 μg/m³) and 64.98% (10.99 ± 4.30 μg/m³), respectively, of the total concentrations during haze days and non-haze days. Aromatics were another major component. They accounted

for 17.32% (10.24 $\pm$ 5.09 µg/m³) and 15.49% (2.62 $\pm$ 1.79 µg/m³), respectively, of the total concentrations during haze days and non-haze days. Alkenes accounted for 14.53% (8.59 $\pm$ 8.05 µg/m³) and 11.85% (2.00 $\pm$ 1.22 µg/m³), respectively, of the total concentrations during haze days and non-haze days.

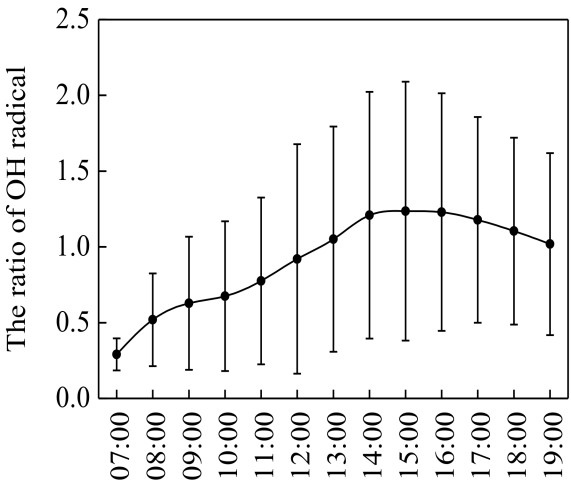

**Figure 2.** Diurnal variation of OH radical concentration ratios during haze days and non-haze days for the daytime.

**Table 1.** Average mass concentrations $\pm$ standard deviation (SD) of 59 VOC species during the observation period at the Chinese Research Academy of Environmental Sciences (CRAES) site in Beijing (µg/m³).

| Variety | Species | Haze days | | non-Haze Days | |
|---|---|---|---|---|---|
| | | Average $\pm$ SD | Proportion | Average $\pm$ SD | Proportion |
| Alkanes | ethane | 9.40 $\pm$ 3.55 | 15.90% | 4.34 $\pm$ 1.06 | 25.67% |
| | propane | 12.40 $\pm$ 7.40 | 20.97% | 3.30 $\pm$ 1.48 | 19.52% |
| | isobutane | 2.79 $\pm$ 1.61 | 4.72% | 0.54 $\pm$ 0.36 | 3.19% |
| | n-butane | 4.88 $\pm$ 2.74 | 8.25% | 1.16 $\pm$ 0.92 | 6.86% |
| | cyclopentane | 1.82 $\pm$ 1.15 | 3.08% | 0.36 $\pm$ 0.32 | 2.13% |
| | isopentane | 1.85 $\pm$ 1.28 | 3.13% | 0.84 $\pm$ 0.63 | 4.97% |
| | n-pentane | 0.03 $\pm$ 0.04 | 0.05% | 0.01 $\pm$ 0.02 | 0.06% |
| | methylcyclopentane | 0.01 $\pm$ 0.04 | 0.02% | 0.01 $\pm$ 0.04 | 0.06% |
| | 2,3-dimethylbutane | 0.01 $\pm$ 0.01 | 0.02% | 0.01 $\pm$ 0.02 | 0.06% |
| | 2&3-methylpentane | 0.00 $\pm$ 0.00 | 0.00% | 0.00 $\pm$ 0.00 | 0.00% |
| | n-hexane | 1.10 $\pm$ 0.86 | 1.86% | 0.18 $\pm$ 0.26 | 1.06% |
| | 2,2-dimethylbutane | 0.02 $\pm$ 0.04 | 0.03% | 0.01 $\pm$ 0.01 | 0.06% |
| | cyclohexane | 0.20 $\pm$ 0.17 | 0.34% | 0.01 $\pm$ 0.04 | 0.06% |
| | 2,3-dimethylpentane | 0.34 $\pm$ 0.30 | 0.58% | 0.03 $\pm$ 0.09 | 0.18% |
| | 3-methyhexane | 0.24 $\pm$ 0.18 | 0.41% | 0.03 $\pm$ 0.05 | 0.18% |
| | 2,2,4-trimethylpentane | 0.04 $\pm$ 0.03 | 0.07% | 0.01 $\pm$ 0.03 | 0.06% |
| | n-heptane | 0.35 $\pm$ 0.32 | 0.59% | 0.03 $\pm$ 0.06 | 0.18% |
| | methylcyclohexane | 0.19 $\pm$ 0.20 | 0.32% | 0.01 $\pm$ 0.03 | 0.06% |
| | 2,3,4-trimethylpentane | 0.01 $\pm$ 0.01 | 0.02% | 0.01 $\pm$ 0.02 | 0.06% |
| | 2-methylheptane | 0.05 $\pm$ 0.06 | 0.08% | 0.01 $\pm$ 0.03 | 0.06% |
| | 3-methylheptane | 0.01 $\pm$ 0.04 | 0.02% | 0.01 $\pm$ 0.01 | 0.06% |
| | n-octane | 0.36 $\pm$ 0.40 | 0.61% | 0.04 $\pm$ 0.05 | 0.24% |
| | n-nonane | 0.14 $\pm$ 0.13 | 0.24% | 0.02 $\pm$ 0.05 | 0.12% |
| | n-decane | 0.01 $\pm$ 0.01 | 0.02% | 0.00 $\pm$ 0.00 | 0.00% |
| | n-undecane | 0.01 $\pm$ 0.04 | 0.02% | 0.01 $\pm$ 0.08 | 0.06% |
| | n-dodecane | 0.13 $\pm$ 0.34 | 0.22% | 0.01 $\pm$ 0.06 | 0.06% |
| total alkanes | | 36.39 $\pm$ 18.64 | 61.54% | 10.99 $\pm$ 4.30 | 64.99% |

**Table 1.** *Cont.*

| Variety | Species | Haze days | | non-Haze Days | |
|---|---|---|---|---|---|
| | | Average $\pm$ SD | Proportion | Average $\pm$ SD | Proportion |
| Alkenes | ethylene | 6.37 $\pm$ 4.69 | 10.77% | 1.68 $\pm$ 0.93 | 9.93% |
| | propene | 0.57 $\pm$ 0.41 | 0.96% | 0.07 $\pm$ 0.12 | 0.41% |
| | trans-2-butene | 0.17 $\pm$ 0.15 | 0.29% | 0.01 $\pm$ 0.05 | 0.06% |
| | 1-butene | 0.89 $\pm$ 0.61 | 1.51% | 0.03 $\pm$ 0.07 | 0.18% |
| | cis-2- butene | 0.22 $\pm$ 0.30 | 0.37% | 0.06 $\pm$ 0.14 | 0.35% |
| | 1,3- butadiene | 0.06 $\pm$ 0.10 | 0.10% | 0.01 $\pm$ 0.05 | 0.06% |
| | trans-2-pentene | 0.03 $\pm$ 0.09 | 0.05% | 0.01 $\pm$ 0.04 | 0.06% |
| | 2-methyl-2-butene | 0.00 $\pm$ 0.00 | 0.00% | 0.00 $\pm$ 0.00 | 0.00% |
| | 1- pentene | 0.01 $\pm$ 0.04 | 0.02% | 0.01 $\pm$ 0.03 | 0.06% |
| | cis-2- pentene | 0.13 $\pm$ 0.14 | 0.22% | 0.04 $\pm$ 0.18 | 0.24% |
| | isoprene | 0.02 $\pm$ 0.03 | 0.03% | 0.01 $\pm$ 0.03 | 0.06% |
| | 2-methyl-1-pentene | 0.06 $\pm$ 0.08 | 0.10% | 0.01 $\pm$ 0.03 | 0.06% |
| | $\alpha$-pinene | 0.01 $\pm$ 0.02 | 0.02% | 0.00 $\pm$ 0.00 | 0.00% |
| | $\beta$- pinene | 0.05 $\pm$ 0.07 | 0.08% | 0.06 $\pm$ 0.08 | 0.35% |
| | limomene | 0.00 $\pm$ 0.00 | 0.00% | 0.00 $\pm$ 0.00 | 0.00% |
| | total alkenes | 8.59 $\pm$ 8.05 | 14.53% | 2.00 $\pm$ 1.22 | 11.83% |
| Alkyne | acetylene | 3.91 $\pm$ 2.65 | 6.61% | 1.30 $\pm$ 1.07 | 7.69% |
| | benzene | 4.18 $\pm$ 2.17 | 7.07% | 0.64 $\pm$ 0.46 | 3.78% |
| | toluene | 3.17 $\pm$ 2.62 | 5.36% | 0.72 $\pm$ 0.52 | 4.26% |
| | ethylbenzene | 0.56 $\pm$ 0.54 | 0.95% | 0.13 $\pm$ 0.17 | 0.77% |
| | m-xylene + p-xylene | 0.59 $\pm$ 0.45 | 1.00% | 0.46 $\pm$ 0.46 | 2.72% |
| | styrene | 0.62 $\pm$ 0.60 | 1.05% | 0.28 $\pm$ 0.24 | 1.66% |
| | o-xylene | 0.01 $\pm$ 0.02 | 0.02% | 0.12 $\pm$ 0.20 | 0.71% |
| | i-propylbenzene | 0.01 $\pm$ 0.01 | 0.02% | 0.00 $\pm$ 0.00 | 0.00% |
| Aromatics | n-propylbenzene | 0.34 $\pm$ 0.30 | 0.58% | 0.05 $\pm$ 0.06 | 0.30% |
| | m-ethyltoluene | 0.04 $\pm$ 0.05 | 0.07% | 0.02 $\pm$ 0.04 | 0.12% |
| | p-ethyltoluene | 0.09 $\pm$ 0.13 | 0.15% | 0.01 $\pm$ 0.03 | 0.06% |
| | 1,3,5-trimethylbenzene | 0.01 $\pm$ 0.02 | 0.02% | 0.01 $\pm$ 0.02 | 0.06% |
| | o-ethyltoluene | 0.11 $\pm$ 0.12 | 0.19% | 0.01 $\pm$ 0.02 | 0.06% |
| | 1,2,4-trimethylbenzene | 0.38 $\pm$ 0.34 | 0.64% | 0.09 $\pm$ 0.10 | 0.53% |
| | 1,2,3-trimethylbenzene | 0.08 $\pm$ 0.05 | 0.14% | 0.06 $\pm$ 0.03 | 0.35% |
| | m-diethylbenzene | 0.02 $\pm$ 0.05 | 0.03% | 0.01 $\pm$ 0.03 | 0.06% |
| | p-diethylbenzene | 0.03 $\pm$ 0.05 | 0.05% | 0.01 $\pm$ 0.01 | 0.06% |
| | naphtalene | 0.00 $\pm$ 0.00 | 0.00% | 0.00 $\pm$ 0.00 | 0.00% |
| | total aromatics | 10.24 $\pm$ 5.09 | 17.32% | 2.62 $\pm$ 1.79 | 15.49% |
| Total VOCs | | 59.13 $\pm$ 31.08 | 100.00% | 16.91 $\pm$ 7.19 | 100.00% |

During haze days, propane, ethane, and n-butane were the most abundant species of the total 26 alkanes. They contributed 34.08%, 25.83%, and 13.41%, respectively, of the total alkane concentrations. Ethane and propane are markers of natural gas emissions [35–37], n-butane is a marker of vehicle exhaust emissions [38,39], and isopentane is a typical tracer of volatile gasoline [38,39]. Therefore, the combustion sources and gasoline exhaust emissions might be important sources for atmospheric VOCs in Beijing. Ethylene, 1-butene, and propene were the most abundant species of the total 15 alkenes that were analyzed, contributing 74.16%, 10.36%, and 6.64%, respectively, of the total alkene concentrations. Of these, 1-butene and ethylene are tracers of the combustion source [28]. Benzene, toluene, and styrene were the most abundant species of the total 17 aromatic compounds, contributing 40.82%, 30.96%, and 6.05%, respectively. Benzene is from fossil fuel combustion sources [36,40], solvent use sources [40,41], and vehicle exhaust emissions [36,40]; toluene is used as a solvent in furniture, footwear, adhesives, printing, and other industries [36]; and styrene is a tracer emitted by the petrochemical industry [36,42]. Although the sequence was slightly different, the top three abundant species of alkanes, alkenes, and aromatics were the same during non-haze days as those during haze days, and can be considered the dominant VOC species in the urban atmospheric environment of Beijing.

*3.3. Source Apportionment*

　　　The ratio between specific VOCs can reflect information about their sources. The ratio method is used to make a preliminary judgment on the main pollution sources before the source analysis of the receptor model. The ratio of toluene and benzene (T/B) has been widely used as a simple method to evaluate the VOC sources. When the T/B value is close to 2, VOCs in the ambient air mainly come from vehicle exhaust emissions [43–45]. When the T/B value is <2, VOCs from other sources are emitted into the atmosphere in addition to vehicle exhaust emissions [43–45]. When T/B is >2, VOCs mainly come from solvent evaporation [43–45]. This method has been adopted by numerous researchers [43–46]. The average T/B values during haze days and non-haze days were $1.01 \pm 0.64$ ($p < 0.01$, $r = 0.82$) and $0.91 \pm 0.46$ ($p < 0.01$, $r = 0.52$), respectively, in this study. This indicated that there were other sources of VOCs in addition to vehicle exhaust emissions. We selected the ratio of acetylene/benzene (A/B), ethylene/toluene (ET/T), and A/ET to discuss the importance of vehicle exhaust emissions during each observation period [47]. The average A/B values during haze days and non-haze days were $2.05 \pm 1.61$ ($p < 0.01$, $r = 0.87$) and $1.14 \pm 0.88$ ($p < 0.01$, $r = 0.63$), respectively, in this study. The average ET/T values in this study were $5.86 \pm 4.17$ ($p < 0.01$, $r = 0.41$) and $4.98 \pm 4.03$ ($p < 0.01$, $r = 0.21$), respectively, and the average A/ET values in this study were $0.74 \pm 0.28$ ($p < 0.01$, $r = 0.83$) and $0.61 \pm 0.21$ ($p < 0.01$, $r = 0.56$), respectively. Overall, the ratio of specific VOC species increased significantly during haze days, indicating that the relative importance of gasoline exhaust emissions during haze days increased.

　　　Six factors were identified during the observation period. These factors were identified as diesel exhaust, combustion, gasoline evaporation, solvent usage, gasoline exhaust, and the petrochemical industry (Figure 3). Factor 1 explained 9.93% of the contribution (Figure 3) and had high loadings of n-nonane (60.35%), n-heptane (25.92%), and 2,3-dimethylpentane (13.22%). The percentages of C3-C5 alkanes, such as propane (8.29%), n-butane (9.50%), and isopentane (16.46%), occupied certain proportions. Benzene, toluene, ethylbenzene, m-xylene + p-xylene, and o-xylene (BTEX) were also evident. Heavy hydrocarbons above C8 are considered as the exhaust gas emission marker of diesel engines [24,36,48]. n-butane is a marker of vehicle exhaust emissions [38,39]. Therefore, Factor 1 was responsible for the diesel exhaust emissions. Factor 2 explained 25.29% of the contribution (Figure 3) and had high loadings of 1-butene (90.47%), acetylene (88.60%), ethylene (51.98%), and ethane (36.28%). Propane, benzene, and toluene were evident in certain proportions. Acetylene,1-butene, and ethylene are tracers of combustion sources [35–37,41]. Propane and ethane are markers of natural gas emissions [35–37]. Therefore, Factor 2 was responsible for the fossil fuel combustion sources. Factor 3 explained 3.90% of the contribution (Figure 3) and had high loadings of C4–C7 alkanes, especially isopentane (58.00%), cyclopentane (58.81%), and cyclohexane (23.40%). Furthermore, 2,3-dimethylpentane and n-butane were evident in certain proportions. Isopentane is a typical tracer of volatile gasoline [23,49], 2,3-dimethylpentane is a tracer emitted by the oil industry [36,42], and n-butane is a vehicle exhaust emission marker [38,39]. Therefore, Factor 3 was responsible for the source of gasoline evaporation. Factor 4 explained 16.88% of the contribution (Figure 3) and had high loadings of ethylbenzene (89.74%), m-xylene + p-xylene (75.54%), o-xylene (58.52%), toluene (41.64%), and m-ethyltoluene (47.52%). Ethylbenzene, m-xylene + p-xylene, and o-xylene mainly come from the use of paints, synthetic spices, adhesives, and cleaning agents [23,36,40]. M-ethyltoluene is produced during the coating process in the metal surface treatment industry [41]. Toluene as a solvent is used in furniture, footwear, adhesives, printing, and other industries [36]. Therefore, Factor 4 was responsible for the source of solvent usage. Factor 5 explained 35.59% of the contribution (Figure 3) and had high loadings of C2–C7 alkanes, such as ethane (17.23%), propane (18.73%), isobutane (58.61%), n-butane (60.34%) (vehicle exhaust emissions marker), and 2,3-dimethylpentane (47.94%). Ethylene and BTEX were found. Therefore, Factor 5 was responsible for gasoline exhaust emissions. Factor 6 explained 8.41% of the contribution (Figure 3) and had high loadings of styrene (79.30%), n-heptane (66.72%), cyclohexane (49.02%), and n-nonane

(24.67%). Cyclohexane is widely used by chemical companies that produce adipic acid and caprolactam [9,50]. N-nonane, n-decane, and n-undecane are important indicators of diesel vehicle exhaust [36,43,44], but may also be released from asphalt and supplied as the raw materials for various oil refining processes [9,50]. Therefore, Factor 6 was responsible for the petrochemical industry.

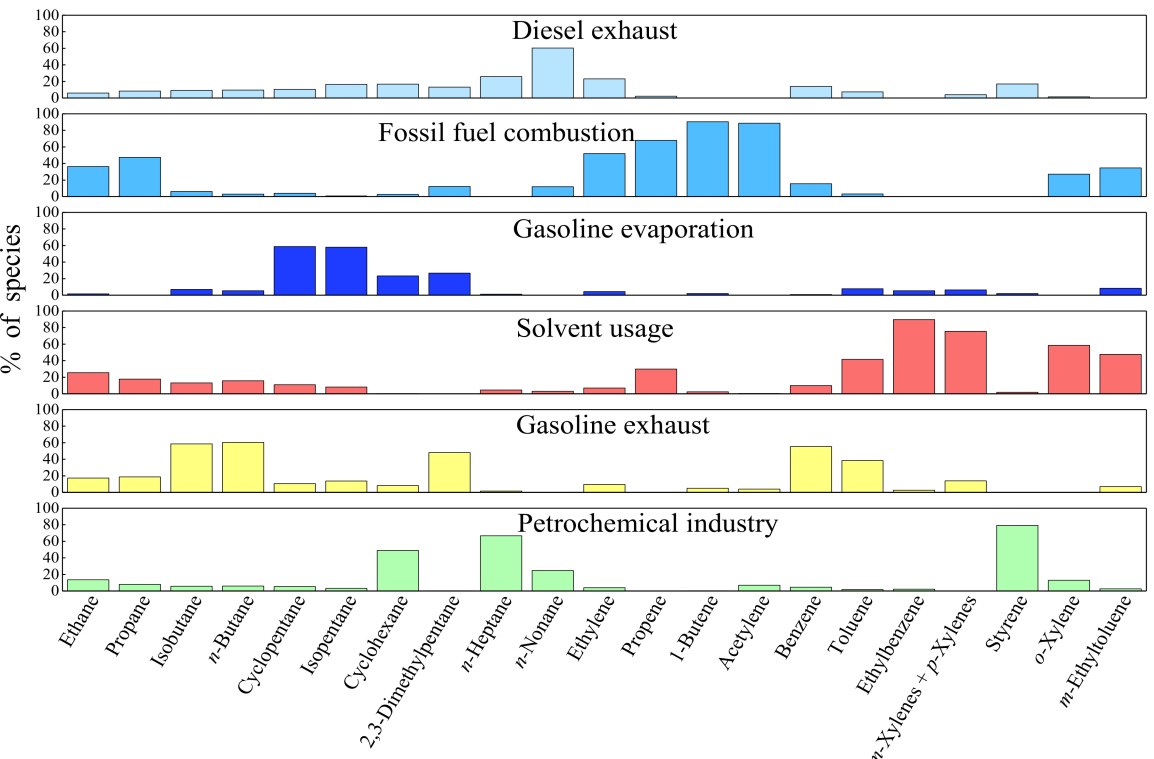

**Figure 3.** Factor profiles (% of species) of each source during the whole observation period.

During the whole observation period (Figure 4), the results suggest that the vehicle exhaust emission sources (gasoline exhaust 35.59%, diesel exhaust 9.93%) were the largest contributor, accounting for 45.52% of the total VOCs. The fossil fuel combustion sources contributed 25.29%. This may be related to the use of coal for heating in northern China. Solvent usage and the petrochemical industry accounted for 16.88% and 8.41% of the total VOCs, respectively. The contribution of gasoline evaporation was the lowest, and was 3.90%.

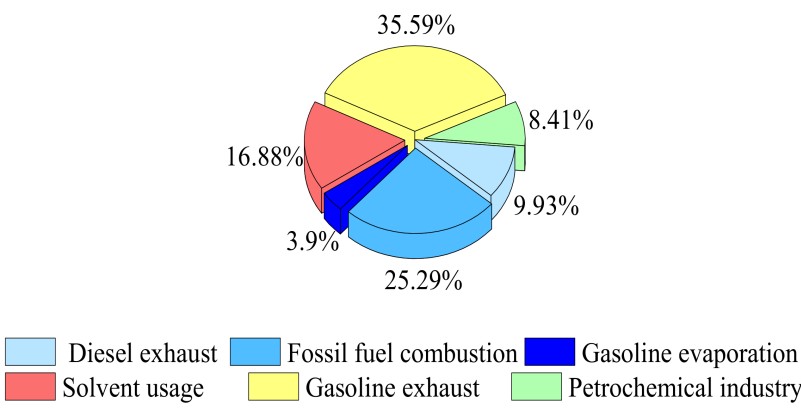

**Figure 4.** The proportion of each source during the whole observation period.

Figure 5 shows the contribution of each source during different pollution events. The contributions of diesel exhaust and the petrochemical industry decreased from 26.14% and 6.43% during non-haze days to 13.70% and 2.75%, respectively, during haze days. The decrease in diesel exhaust and the petrochemical industry can be explained by numerous emergency measures implemented by the government during the haze events, such as shutting down highly polluting enterprises and banned diesel vehicles, which were aimed at alleviating the haze pollution [10]. The contributions of solvent usage and gasoline evaporation did not prominently change during different pollution events, which showed that these were not the main factors that aggravated the VOC pollution. However, the gasoline exhaust contributed 48.77% during haze days, which further implied that the contribution of gasoline exhaust emissions increased during haze days.

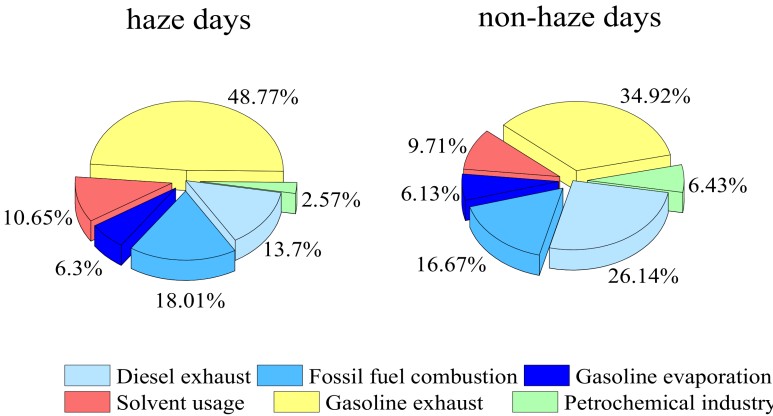

**Figure 5.** The proportion of each source during different pollution events.

### 3.4. The Potential Source-Areas of VOCs

Backward trajectory cluster analysis was used to reveal the transport pathways of air masses. Figure 6 shows the 24 h backward trajectories from CRAES during haze and non-haze days. The dominant air masses of CRAES and the proportion of VOC species of each air mass trajectory during haze and non-haze days were observed. The air mass of VOCs during haze days was mainly affected by the short-distance transportation from the southwest of Hebei province (35.8%) and the local area centered in Beijing (22.5%), which were the most important pollution trajectories. However, the air mass of VOCs during non-haze days was mainly affected by long-distance transport. The long-distance air masses of the non-haze days came from the northwest, accounting for 53.3% and 34.2%, respectively. Regardless of the distinction between haze and non-haze days, other air masses of trajectories have a relatively small proportion. Regarding the proportion of VOCs in each air mass trajectory, it could be seen that alkanes contributed more than half of all air masses. However, the contributions of alkenes were larger than those of aromatics in northwest transport pathways during haze days, whereas the contributions of aromatics were larger than those of alkenes in southwest transport pathways during haze days and in all transport pathways during non-haze days. Overall, the main influences were the short-distance transport of air masses during haze days and the long-distance transport of air masses during non-haze days, which had more alkenes than aromatics.

The source regions of VOCs were identified by the potential source contribution function (PSCF) and concentration-weighted trajectory (CWT) models in this study. A higher PSCF value represents a region with a higher probability as a source area for VOC pollution, and a higher CWT value means that the region has a higher contribution to VOC concentrations [51,52]. Figure 7 shows the PSCF (left) and CWT (right) from CRAES during haze and non-haze days. The source area results identified by the CWT model in Figure 6 (right) were very similar to those analyzed by the PSCF model in Figure 7 (left). For haze days, the area with high PSCF values were the southwest of Hebei province

and the local area centered in Beijing, whereas the PSCF values of the northwest areas were lower (Figure 7 (left)). The corresponding backward trajectories from the southwest were short and moved slowly, so air pollutants easily accumulated and caused severe air pollution. Fast-moving air masses from the northwest, involving long-distance air transport carrying clean air masses, tended to disperse VOC concentrations (Figure 7 (right)) [52]. Interestingly, the areas with high or low PSCF and CWT values were all located to the northwest of Beijing during non-haze days. Many heavy polluted factories are located in the southwest and south of Hebei province. Beijing readily accumulates air pollutants from the southwestern and southern regions during stagnant weather, causing the concentration of atmospheric pollutants to increase rapidly. Thus, regional cooperation for control of industry emissions is essential during haze days, particularly in the cities to the southwest and south of Beijing.

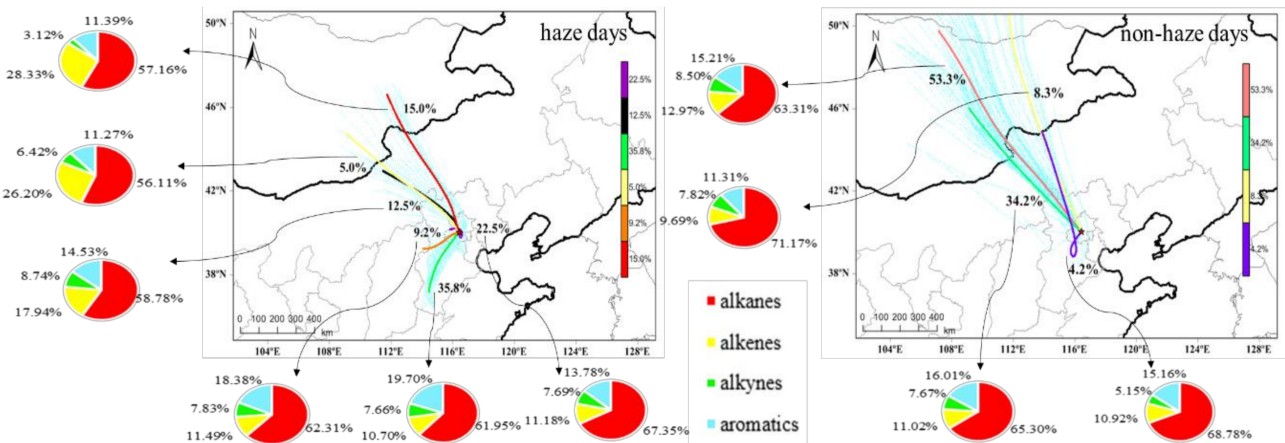

**Figure 6.** The 24 h backward trajectories from CRAES during haze and non-haze days.

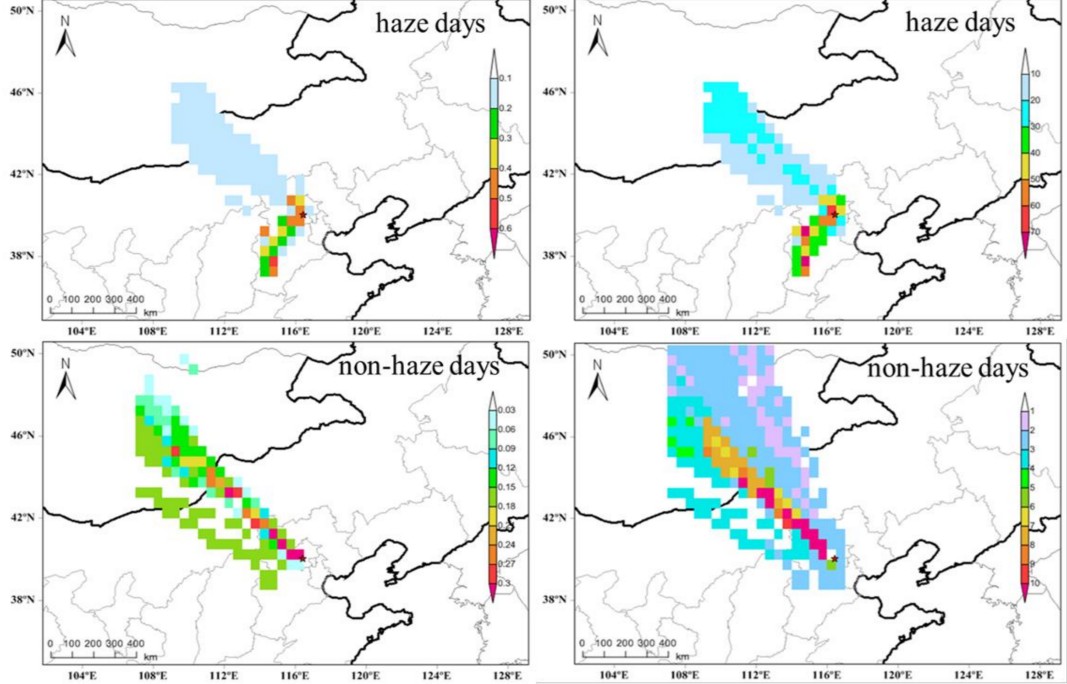

**Figure 7.** The potential source contribution function (PSCF) (**left**) and concentration weighted trajectory (CWT) (**right**) from CRAES during haze days and non-haze days.

## 4. Conclusions

In this study, continuous observation of 59 VOCs species was carried out in the urban area of Beijing using an Airmo VOC (GC-866) online analyzer during 1–15 March 2019. The increase of the VOC concentrations during haze days in Beijing were mainly ascribed to the static stability caused by lower wind speed. The decreased average of $O_3$ concentrations during haze days were mainly ascribed to the extremely low concentrations at night during the observation period instead of the decrease of photochemical reactions of VOCs as reported. The average OH radical concentration during haze days were higher than those of non-haze days from 13:00–19:00 during 1–15 March, indicating that the rate of photochemical reaction during haze days was higher than that of non-haze days. The proportions of diesel exhaust and the petrochemical industry emissions decreased significantly during haze days due to the implementation of numerous emergency measures by the government, whereas the contributions of gasoline exhaust emissions increased during haze days. The backward trajectories, PSCF and CWT showed that the air mass of VOCs during haze days was mainly affected by the short-distance transportation from the southwest of Hebei province; however, the air mass of VOCs during non-haze days was mainly affected by the long-distance transport from the northwest.

**Author Contributions:** All authors made significant contributions to this study. L.Z. analyzed data and wrote the paper. Y.Z. provided structure and data analysis methods, and reviewed and revised the paper. X.W. reviewed and revised the paper. H.L. and K.Z. shared ideas about the work. N.C. and L.L. provided guidance for the PSCF and CWT models. All authors agree to the submission of this paper. All authors have read and agreed to the published version of the manuscript.

**Funding:** This study was supported by Basic Scientific Research Fund in Nation Nonprofit Institutes, China (No. 2016-YSKY-026), and National Research Program for Key Issues in Air Pollution Control (No. DQGG0304-05).

**Acknowledgments:** We sincerely thank Junyi Lv from Shanghai Thunder Environmental Technology CO., Ltd. and Chenfei Liu from the Chinese Research Academy of Environmental Sciences for helping us. They were involved in instrument maintenance and calibration. Finally, we would like to express our immense gratitude to the reviewers and editors who have contributed valuable comments that helped us to improve the quality of the paper.

**Conflicts of Interest:** All authors declare that there is no conflict of interest.

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
