# Peer review of "Variations in Levels and Sources of Atmospheric VOCs during the Continuous Haze and Non-Haze Episodes in the Urban Area of Beijing: A Case Study in Spring of 2019"

_atmosphere, doi:10.3390/atmos12020171_

Round 1

Reviewer 1 Report

General comments:

This manuscript applied the observational data and the backward trajectory method to study the VOC emission source. The ideas in this manuscript are good. However, some of the result parts are not clear (ozone formation, ozone level) and miss some critical information (OH radical calculation). I suggest the authors should edit those parts again or remove those parts before publishing.

Specific Comments:

  1. In Equation 3, the EA/A should be ET/A.
  2. I suggest that the paragraph in Line 184-208 should be re-edit when explaining the air pollutants pattern during the measurement period. Here are my reasons:
    a). Because the positive net ozone formation process only happened in the daytime, the ozone pollution should consider two indicators: maximum daily one-hour average ozone and maximum daily 8-hour average (MDA8) ozone. In Figure 1, the peak hourly ozone during haze days (3/1-3/5) is 120-160 ug/m3. The one-hour ozone during haze days is higher than the one-hour ozone on non-haze days. Therefore, lines 184 and 185: “The diurnal variation of ozone was completely different” and “ Unlike the concentrations of other pollutants…“ are not appropriate.
    b). Line 194-196 didn’t explain the real cause and effect of the night-time ozone pattern during haze day and non-haze day. The night-time ozone has different chemical reactions, including NO2+O3 -> NO3, NO+O3-> NO2 and O3 + Alkene-> OH radical
    At least, the author should mention the NO emission and explain the NO titration process at night (NO+O3).
    c). Why NOx concentration become almost zero on 3/6 and 3/7?
    d). Why is PM2.5 higher at night than in the daytime? The author should discuss the PBL height process in the description and lower PBL cause the pollutants to accumulate.
    e). Line 197-199 is hard to understand, and the idea is incorrect. I think haze days have higher MDA1 and MDA8 ozone.
    f). Line 205-208, please present the hourly OH radical ratio for the daytime hour only and show the result for haze day and non-haze day to support this idea.
  3. The wind speed and wind direction plot in figure 1 should add a scale bar for understanding the wind speed range.
  4. Line 308-309 should add a reference for explaining that the government shut down the human activities and emission processes.
  5. Line 357-359, “The increase of the VOCs concentration……lower OH radical concentration”. I didn’t see the OH radical ratio calculation result in the result section. The results of the daytime hourly OH radical ratio between haze day and non-haze day should be presented before the conclusion.

Reviewer 2 Report

General comments

The authors analyze the atmospheric composition and the role of different source categories during haze and non-haze episodes in Beijing, China. To this aim, they use different tools, such as positive matrix factorization, back-trajectories, potential source contribution function and concentration weighted trajectory.

My greatest concern about this work is that it is based on a very short period of observation, only 15 days. Within this period there are only two haze events (and two non-haze), as shown in figure 1. Are two events enough to get results of general validity?

Another general comment is about the English language: it must be improved in the whole manuscript.

Other minor comments are listed below.

Detailed comments

  • Line 68: Add reference number for "Sun".

  • Lines 74-77. Review the sentence.

  • Line 87: “detailed information”, not “detail information”

  • Lines 88-89: Review the sentence. According to the abstract the observation period should be 15 days long. I guess 8 days were haze, and 7 days were non-haze, but this is not clear from the sentence.

  • Line 89: Remove one of the two "average".

  • Line 91: Not sure if this date format is accepted. You should format dates as YYYY/MM/DD, or YYYY-MM-DD.

  • Line 95: “detailed information”, not “detail information”

  • Line 96: From "External" to "0.9". Review the sentence, at least a verb is needed.

  • Lines 97-98: It is not clear if a deviation smaller that 10% was a requirement or a result.

  • Line 100: “detailed information”, not “detail information”

  • Line 104: "receptor model" after (PMF) could be deleted.

  • Paragraph 2.3.1: The whole paragraph must be rewritten. It is currently written more in the form of personal notes than in a form suitable for a scientific paper. For example, Equation 1: I guess concentration is the measured value when it is greater than MDL, and MDL/2 otherwise. A sentence must be added to explain this. Also, Points 1 and 2: I guess they are indications about how to select the values of EF. Sentences must be added to explain it. The notes are not quite clear.

  • Line 119: Add a reference, or a link for the PMF 5.0 software (e.g., https://www.epa.gov/air-research/positive-matrix-factorization-model-environmental-data-analyses)

  • Equation 3. "EA" in the equation should be "ET".

  • Line 129: "represent" not "represents"

  • Line 133: Reference [33] should be added for TrajStat.

  • Line 140: It should be "combining", not "combined".

  • Line 141: "identifying locate" or "identifying the location of"?

  • Line 165: “parameters”, not “parameter”

  • Line 172: "While" does not seem to be correct at the beginning of this sentence. One would expect something similar to "While it looks warm outside, there is actually a cold breeze so it is not warm at all", but the second part is missing. So I believe "While" must be deleted.

  • Line 184: "appeared" or "appearing"?

  • Lines 193-195: From "O3" to "values". The behavior of NOX and O3 is clear in figure 1, but the sentence is not very clear. Please try to improve it.

  • Line 197: "the nearly" should be "nearly", without any article.

  • Line 198: Use "observation" in place of "observe".

  • Figure 1: Add a scale for the arrows representing the wind speed. A horizontal arrow is already present, just write the number (in m/s) corresponding to its length.

  • Line 210: “parameters”, not “parameter”

  • Line 226: "tracer", not "tracers".

  • Line 242: "their source information" or "information about their sources"?

  • Line 257: “days”, not “day”

  • Line 271: “sources”, not “source”

  • Line 280: Delete one of the two "paints".

  • Line 281: Is "synthetic spices" correct? If yes, please add few examples.

  • Line 313: "important", not "importance"

  • Lines 326-328: From "The" to "days". Review the whole sentence.

  • Figure 5: The aspect ratio of the figure does not seem correct. Circles are actually ovals.

  • Lines 378-379: It is a bit strange to get acknowledgments before doing the review. Anyway, thanks.